# Nonclassical Systemics of Quasicoherence: From Formal Properties to Representations of Generative Mechanisms. A Conceptual Introduction to a Paradigm-Shift

**Gianfranco Minati**

Italian Systems Society, 20161 Milan, Italy; gianfranco.minati@airs.it; Tel.: +39-02-6620-2417

**Abstract:** In this article, we consider how formal models and properties of emergence, e.g., long-range correlations, power laws, and self-similarity are usually platonically considered to represent the essence of the phenomenon, more specifically, their acquired properties, e.g., coherence, and not their generative mechanisms. Properties are assumed to explain, rather than represent, real processes of emergence. Conversely, real phenomenological processes are intended to be approximations or degenerations of their essence. By contrast, here, we consider the essence as a simplification of the phenomenological complexity. It is assumed to be acceptable that such simplification neglects several aspects (e.g., incompleteness, inhomogeneities, instabilities, irregularities, and variations) of real phenomena in return for analytical tractability. Within this context, such a trade-off is a kind of reductionism when dealing with complex phenomena. Methodologically, we propose a paradigmatic change for systems science equivalent to the one that occurred in Physics from object to field, namely, a change from interactional entities to domains intended as extensions of fields, or multiple fields, as it were. The reason to introduce such a paradigm shift is to make nonidealist approaches suitable for dealing with more realistic quasicoherence, when the coherence does not consistently apply to all the composing entities, but rather, different forms of coherence apply. As a typical general interdisciplinary case, we focus on so-called collective behaviors. The goal of this paper is to introduce the concepts of domain and selection mechanisms which are suitable to represent the generative mechanisms of quasicoherence of collective behavior. Domains are established by self-tracking entities such as financial or are effectively GPS-detectable. Such domains allow the profiling of collective behavior. Selection mechanisms are based on learning techniques or cognitive approaches for social systems.

**Keywords:** coherence; collective; cognitive; domain; dual; field; incompleteness; quasi; selection mechanism

---

## 1. Introduction

In the literature, the difference between noncomplex and complex systems may be seen as running parallel to the distinction between Bertalanffy- and post-Bertalanffy systemics approaches [1–3].

However, systemics approaches commonly consider the problems and properties of systems in a classical way, that is, a way which is representable according to interaction-based, formal, complete models, and conceptual approaches of classical physics, in particular, nonrelativistic and nonquantum approaches. The pinnacle of nontraditionality is achieved by considering nonlinear aspects of complexity such as chaoticity, self-similarity, and network properties.

However, we think that some unsolved problems of systemics approaches may be very good entry points to delineate new options to represent systems and their generative mechanisms through which properties different from those associated with composing elements are acquired.

Examples of unsolved problems include the unavailability of robust approaches which are suitable to noninvasively act on coherent, complex systems such as collective behaviors, e.g., avoiding, inducing, orientating, deactivating, and modifying their coherence; dealing with quasisystems ([4], pp. 102–128, [5]), i.e., when a system is not always a system and not always the same system; dealing with different levels and types of coherence; dealing with systems whose components are in large numbers (potentially tending to infinity), mutating and unstable; and when coherences occur in a nonconstant way but through sequences of partial coherencies that are to be resumed (systems are locally incoherence-tolerant). We consider the current systemics methods to be useful for representing, rather than explaining, complex phenomenological systems, dealing instead with well-defined abstract systems, i.e., often formal, sophisticated toy-systems. This is particularly the case of collective beings [6], i.e., coherent emergent systems whose components are provided with cognitive systems allowing decisions to occur, rather than physical behaviors. This makes current systemics methods unsuitable for real problems of human systems such as in economics, ecosystems, the environment, finance, medicine, and social systems. It is a matter of finding systemics approaches that are compatible with theoretical incompleteness for nonreductionistic science, i.e., the science of emergence.

Formal models are usually platonically considered to capture the essence of real phenomenological processes and are intended as material approximations or degenerations of their essence. Neglected aspects are assumed to be irrelevant and insignificant in comparison to the predominant essence.

It is assumed to be acceptable that such simplification neglects several aspects (e.g., incompleteness, inhomogeneities, instabilities, irregularities, and variations) in return for analytical tractability. This applies to noncomplex phenomena and noncomplex systems, i.e., where no processes of emergence occur. It is usually viewed as an acceptable reductionist approach.

Furthermore, the focus is on the essence of the acquired properties, such as coherence, paying little attention to real generative processes. All of this is assumed to be generalization.

In this latter case, the usage of essence-based approaches is a form of new reductionism.

By contrast, here, we consider the essence as an unsuitable simplification of the phenomenological complexity where multiplicity, incompleteness, and singularities of quasicoherence have predominant, unignorable roles in representing phenomenological processes by acting as explications rather than the properties of representations, e.g., nonlinearity.

Here, we consider that such a trade-off is unacceptable when dealing with the phenomenological complexity of several processes such as in economics, medicine, and social systems, i.e., where processes of emergence take place and their real mechanisms cannot be neglected.

We mention the increasing availability of related new mathematical approaches, such as the mathematics of nature-inspired computation, networks, and subsymbolic approaches.

As a typical general interdisciplinary case, we focus on so-called collective behaviors, self-organization, and emergence as conceptual general frameworks where, typically, aspects such as incompleteness, temporal incoherencies, incoherence-tolerance, multiplicities, structural dynamics (systems are not always systems and are not always equal to themselves), and mutations, e.g., phase-transition-like and radical emergence, apply in the face of identity preservation, and coherence-resumption.

Methodologically, we conceptually propose that collective behavior no longer be considered as generated by the behavioral, microscopic properties of interacting entities, but rather, as compliance with domains, an extension of the concept of field which, in physics, since electromagnetism, has replaced the classical understanding of objects. Domains are characterized by the availability of multiple simultaneous options, types of multiple fields, and are considered both as generators and as being generated by collective behaviors.

The purpose of this article is to introduce some conceptual proposals outlining possible alternative understandings to the standard approach based on entity interaction, which are, rather, based on the concept of domain and selection mechanisms that are suitable for dealing with unsolved problems of systemics, giving rise to less abstract and more practical approaches.

In Section 2, we shortly outline the classic entity-based approaches used to introduce, model, and deal with systems. In particular, we mention the approaches considered by Von Bertalanffy and Klir as representative of still generic understandings of systems. We methodologically mention the related paradigm shift that we propose in this article from the usual collective interaction-based approaches to domains of options and the related selection mechanisms. Domains are established by self-tracking entities, e.g., financial, effectively GPS-detectable. Such domains allow the profiling of collective behaviors. Selection mechanisms are based on learning techniques and cognitive approaches for social systems.

In Section 3, we consider some introductory concepts that are suitable for subsequent proposals. In particular, we distinguish among autonomous and nonautonomous entities (respectively provided or not provided with cognitive systems), and we specify generic, local, and global constraints to be considered for the dynamics of single entities as elementary cases. The more sophisticated cases relate to populations of collective constraints.

In Section 4, we provide a short overview of microscopic-based approaches to model collective interaction-based behaviors and properties of their coherence.

In Section 5, we provide a short overview of mesoscopic-based approaches to model collective behaviors, such as those of metastructures, based on properties of clusters and infraclusters of entities.

We start Section 6 by mentioning how, in theoretical physics, after the historical introduction of the duality corpuscular-wave aspects and various properties of quantum physics, e.g., entanglement, fields are considered to be the primary entities rather than objects or sources understood to be equivalent to specific space–temporal places characterized by very strong intense concentrations of a field.

When looking for alternative approaches to classical interaction-based ones among microscopic constituting entities that establish emergent collective behaviors which are modeled using properties such as synchronization, (long-range) correlations, self-similarity, power laws, and networking [7,8], we mention—for completeness, since it is not further elaborated here—the Nambu-Goldstone bosons (NGBs) in quantum physics. In the quantum vacuum, each perturbation causes the emergence of collective long-range excitations termed NGBs. They coordinate the behavior of individual components of the system, keeping a general coherence. NGBs are intended to act as "coherence keepers", a role characterizing one of most important aspects of processes of emergence.

After introducing the core proposals of the article, we consider domains conceptually correspondent to fields. However, fields are given by the value assumed by a variable at a specific point in space and time, while the domain of an entity is given by the values of variables that represent the possible dynamics can assume at the following instant.

A generic local domain is assumed to be a virtual multidimensional (dimensions are the degrees of freedom established by rules of achievability, admissibility, compatibility, continuity, equivalence, nonequivalence, probability, and ranges of admissible values in a predefined granularity of time) space of values, virtually generated by single, supposed isolated entities or, more interestingly, by collective entities as interrelated, networked local domains of single entities.

In elementary cases, we consider the domain of single, supposedly isolated entities with local domains.

In sum, the local domain of an entity is its possible authorized dynamics.

In the case of multiple collective entities, we consider their authorized clusters of interrelated (a specific selection reciprocally affects, implies, or prevents other selections for other variables) instantaneous local domains named here configuration domains. Depending on the level of description, clustered local domains relate to correspondent entities such as neighbors or have a limited well-defined topological distance.

In sum, the configuration domain of a cluster of entities is its possible authorized dynamics.

The coherent subsequent sequences of such configuration domains establish a temporally global configuration domain, intended to be a suitable global cluster—in this case, networked—of temporally subsequent configuration domains. The global configuration domain is assumed to correspond to the phenomenological dynamics of collective behavior. We stress that when dealing with phenomenological dynamics, configuration domains may not necessarily be coherent or may have temporal, occasional local coherences that are different between them and the global one. As we shall see, it is matter of quasicoherence that must be continuously established and restored, rather than phenomenologically improbable formal coherence. The general domain is assumed to represent the subsequent changes that are possible for the entire collective entity, i.e., the possible dynamics for collective behavior.

In sum, the global configuration domain of the collective behavior of entities represents its possible authorized dynamics.

As we shall see, the coherent dynamics of the collective behavior is conceptually intended due to a process of choice as selection performed by composing entities among authorized domains (among roles, see Section 7) through cognitive selection for collective beings and a selection mechanism, for instance, energy-based in the case of nonautonomous entities.

We discuss the properties of domains and how the handling of such properties may be considered a suitable approach to act on the corresponding collective behavior, for instance, to categorize, influence, induce and avoid them. Examples are the network properties of the domains, including topological properties, related to stability, continuity, and oscillatory properties.

We present effective approaches to identify domains for nonautonomous entities and as cognitive domains for autonomous entities. We present effective approaches to represent selection mechanisms as learning and analog computational processes.

In Section 7, we extend the concept of domain by considering domains of roles for composing entities of collective systems. A role for a generic entity is intended as a simultaneous assumption at time $t_n$, of specific values among the ones authorized by the domains. A role should be intended to represent a specific usage of the domain, represented by a vector of specific values or ranges of possible values among the ones authorized by the domain. Such simultaneous vector aggregation of authorized (since belonging to domains) values represents real phenomenological properties—physical for nonautonomous entities, physical and cognitive for autonomous entities.

We particularly discuss the case of collective beings whose cognitive-based decisions are suitable to allow quasicoherence phenomena (see the concept of quasisystems in [4] and theoretical incompleteness in [9]) when coherence may tolerate variable percentages of temporal incoherence.

Tolerance should be measured, for instance, in terms of percentages, variations, periodicity, etc., allowing interesting profiling information.

In Section 8, we discuss the subject of the cognitive generation of coherence in collective beings and its incompleteness and restorability, rather than its formal properties.

Coherence is assumed as a matter of suitable selection among roles.

Coherence is intended among temporal sequences of roles in configuration domains. Microscopic properties of composing entities, such as long-range correlations, synchronization, and self-similarity, are intended here to be suitable to detect and represent rather than to generate and explain coherence.

Furthermore, we consider quasicoherence rather than formal, idealistic coherence.

It is important to consider imperfect, incomplete coherencies, such as those—occurring regularly or irregularly—of percentages of coherent constituting entities, the tolerance for incoherencies, the processes of resuming coherencies, the formation of small-world coherent communities, the processes of changing coherence, e.g., radical emergence, protein folding and acquisition of superconductivity, and superfluidity.

The properties of global configuration domains, such as networks of configuration domains and the related selection mechanisms, can be considered to replace the usual interaction-based mechanisms

of classical systemics methods by conceptually substituting materials composing entities in conceptual correspondence with theoretical physics considering fields rather than objects.

In Section 9, we present some synthesizing remarks about the purposes of the article.

In Section 10, we mention how further research is expected to allow representations of real domains and learned, analogue selection mechanisms to suitably represent and explain significative and predominant aspects of complexity neglected by formal approaches—for instance, through usage of learning, profiling techniques, and the mathematics of natural-inspired computation.

The outlined approach should not be considered in the idea to look for the best, more powerful one, but, rather, for mixed, context-dependent usages, combining, for instance, microscopic, mesoscopic, and domain-based approaches and models, depending on the effective situation and interests of the researcher.

In Section 11, we conclude the article, mentioning how such multidimensional approaches should benefit representations and applications when dealing with real phenomena of emergence and the acquisition of coherence in a disciplinary context where formal models are ineffective to deal with real phenomenological complexity, such as in architecture, finance, medicine, and social systems, e.g., security, health, waste production and management, and resource consumption.

Finally, for the reader's convenience, we have included boxes containing brief, condensed information regarding the issues considered in the chapter.

## 2. Background and Methodology: From Interactional Entities to Domains.

The general subject of this article pertains to the generic processes of transformation of sets of single entities into general entities acquiring properties that single entities do not possess. Examples of properties possessed by classic, i.e., nonquantum, entities are their age, capacity of containers, dimensions, electrical resistance, hardness of materials, heat conductivity, sharpness as for knives, and weight (we do not consider quantum super-conductivity and super-fluidity).

This transformation is performed, for instance, by chemical reactions; by inserting a structure among entities, such as organizing them as machines; or by using physical reactions, for instance, magnetism and lasers.

Such generic transformational processes are the interest of systemics. General entities acquiring properties that single entities do not possess are termed systems, and systems science studies properties of such interdisciplinary processes of transformation and of systems.

The purpose is to identify generic, highly representative systems, categories of systems, and their properties having interdisciplinary or very general disciplinary validity in order to allow reusable and standardizable approaches. This is a fundamental part of science and of systemics where general entities are systems.

### 2.1. Interaction, Entity-Based Systems

This originated from the General Systems Theory that we shall introduce shortly as the background of our proposal.

The mathematical biologist Ludwig von Bertalanffy (1901–1972), the father of *General Systems Theory* [10–13], considered a system as characterized by suitable state variables $Q_i$, $Q_2$, ... , $Q_n$ whose instantaneous values specify the state of the system.

The classical *general* representation of collective (organized, self-organized or emergent, see Section 3) interaction within a system characterized by suitable state variables $Q_1$, $Q_2$, ... , $Q_n$ whose instantaneous values specify the state of the system, is given by their time evolution ruled by a system of ordinary differential equations, such as:

$$\begin{cases} dQ_1/dt = f_1(Q_1, Q_2, \ldots, Q_n) \\ dQ_2/dt = f_2(Q_1, Q_2, \ldots, Q_n) \\ \ldots\ldots\ldots\ldots\ldots\ldots\ldots\ldots\ldots\ldots \\ dQ_n/dt = f_n(Q_1, Q_2, \ldots, Q_n) \end{cases} \tag{1}$$

The system specifies how a change in the value of a given state variable, $Q_n$, is related to all other state variable changes through $f_n$, representing their collective interaction being governed and structured by $f_n$.

$f_n$ represents interdependence and interaction.

Furthermore, forms of indirect interaction are possible when mediated by intermediary entities, perturbing or elaborating the received interaction, for instance, in the form of energy or information.

We mention from the enormous literature on systems the contribution of George Klir (1932–2016), starting from a compilation of definitions of system. He introduced a definition of system in five levels [14–17].

First, Klir introduced some basic definitions:

- *The ST-Structure (State transition)* as "The complete set of states together with the complete set of transitions between the states of the system."
- *The UC-structure (Structure of the universe and couplings)* as "A set of elements together with their permanent behaviours and with a UC characteristic."
- *The space–time resolution level* intended as the feature of the observations or measurements, e.g., quantities to be considered, accuracy of measurements, and frequency.
- *Permanent (real)* behavior as defined by the relationships linking given quantities at the resolution level.

The related five definitions of system introduced are:

- *Definition by a set of external quantities and the resolution level*: The system is a given set of quantities regarded at a certain resolution level.
- *Definition by a given activity*: The system is a given ensemble of variations in time of some quantities under consideration.
- *Definition by permanent behavior*: The system is a given time-invariant relation (even probabilistic) between instantaneous and/or past and/or future values of external quantities.
- *Definition by real UC-structure*: The system is a given set of elements, together with their permanent behaviors, and a set of couplings between the elements, and between the elements and the environment.
- *Definition by real ST-Structure*: The system is a given set of states together with a set of transitions between the states.

Such definitions and other contributions were of great importance, particularly in engineering.

Several other disciplinary approaches have been considered and introduced (for a short overview, see ([4], pp. 2–20; [6], pp. 5–42).

### 2.2. From Entities to Fields, Domains.

The general, dominant, transversal approach is entity-based. This is in contrast with physics, where the discovery of electromagnetism introduced the opportunity to consider the fields (and, subsequently, quantum fields, see Section 6.1) as primary entities rather than objects or sources understandable as equivalent to specific spatiotemporal places having intense concentrations of a field.

We are trying here to methodologically introduce such a paradigmatic change as that from object to field in physics to systemics, as a change from interactional entities to domains intended as extensions of multiple fields, as it were.

We consider as an appropriate case for study the generic phenomenon of emergence of collective behavior as no longer generated by behavioral microscopic interactional properties of entities, but, rather, as due to the dual generation of and compliance with domains, extensions of the concept of field in physics.

Fields prescribe a well-defined value to any entities at a point. Phenomenological dynamics and processes of emergence contemplate the availability of locally equivalent values having, however, different global effects, such as coherence. Different global coherences are possible, maintaining, nevertheless, the collective behavior, e.g., a flock or a swarm. Such equivalence of collective behavior is considered here as due to domains of multiple equivalent values, allowing entities to adopt macroscopic roles, intended as multiple simultaneous assumptions of specific values among the ones allowed.

Domains are metaphorically intended as simultaneous multiple fields, among which the entity, for any reason, such as energetic or cognitive, breaks the equivalence and adopts or chooses values, i.e., roles (see Section 6.1).

The multiplicity in principle of domains considered here can be considered as based on previous approaches dealing with multiplicity in science. Apart from classic interdisciplinarity, we mention how complexity requires multiple, nonequivalent approaches, as stated by the dynamic usage of models—DYSAM ([4], pp. 64–88), and Multiphysics, defined as the coupled processes or systems involving more than one simultaneously occurring physical field, see [18,19].

## 3. Nonautonomous and Autonomous Entities and Constraints

Here, we introduce some specifications by differentiating between nonautonomous and autonomous entities that are supposed to deal differently with generic, global, and local constraints. In the following chapters, they are considered to deal with the emergence of coherence and collective behaviors.

### 3.1. Nonautonomous and Autonomous Entities.

At this point, it is appropriate to distinguish between collective systems established by the collective coherent behaviors of their supposed (depending on the level of representation) constituting composing entities—in the case of nonautonomous entities, from the collective systems established by autonomous entities. In short, an autonomous entity is intended to have a cognitive system, to be understood as a system of interactions among, at different levels, cognitive activities such as those related to attention, perception, language, the affective and emotional sphere, memory, the inferential system, and logical activity.

We distinguish among autonomous entities, supposed to be able to decide, even if at different levels of consciousness, their behavior, and nonautonomous entities, supposed to just be able to react only in an algorithmic, computable way.

Examples of collective systems established by nonautonomous entities, i.e., without or assumed to be without cognitive systems, include amoeba colonies; bacteria colonies; nematic fluids, e.g., liquid crystals; shaken metallic rods, e.g., periodically vertically vibrated granular rods spontaneously form vortices which grow with time [8]; whirlpools in fluid molecular dynamics; protein chains and their withdrawal; coherent chemical reactions where composing molecules assume coherent global behaviors such as the well-known Belousov–Zhabotinski reaction [20,21] consisting of an oscillating chemical reaction in which the periodic variation of concentration is indicated by striking color variations; the formation of convective patterns of molecules called Rayleigh–Bénard cells [22] in a liquid evenly heated from below; objects on vibrating surfaces that tend to make consistent variations; and signals when networked, such as on the Internet.

We may then consider examples of coherent communities of living systems provided with non-negligible cognitive systems.

Examples include anthills; herds; schools; swarms; flocks; industrial clusters; industrial district networks; markets; and social systems, such as cities, schools, hospitals, companies, families, vehicle

traffic, and temporary communities, such as passengers, audiences, and telephone networks. As mentioned above, we distinguished this case by naming such collective systems as collective beings [6].

See Box 1 for a summary.

**Box 1.** Summary No. 1.

---

*Nonautonomous Entities*

These are usually inanimate entities driven only by physical forces, e.g., fields, and their interactions occur through exchanges of energy,

*Autonomous Entities*

These are animate entities provided with a cognitive system of suitable complexity, allowing them to learn, memorize, perceive, decide, and for higher complexity, make logical inferences and have languages.

---

*3.2. Generic, Local, and Global Constrains*

Now, we consider generic constraints for variations in values of variables assumed to be necessary for the emergence of any collective behavior, such as (a) the physical admissibility of the corresponding changes that should respect the general physical evolutionary constraints of the phenomenon (such constraints will take the form of ranges of validity)—for instance, very strong (depending on the scale in use) changes in speed, altitude or direction cannot occur instantaneously but need some graduality; (b) the compatibility intended to be a local property between the values of variables representing the neighborhood along time—for instance, compatibility is assumed to avoid collisions and disproportionality in changes such as in collective motion and market dynamics; and (c) the equivalence of possible authorized values inducing equivalent local or global effects. Such equivalence is related to the unpredictability of processes of emergence when the different options for systems are equivalent and the selections occur in different ways, such as through fluctuations and symmetry breaking [5].

### 3.2.1. Local Constraints

At this point, we should differentiate between local constraints and global constraints.

In the case of nonautonomous entities, examples of local constraints include limits, and ranges of validity for short-range attractive and repulsive forces; short-range energy exchanges, e.g., possible nonhomogeneous friction; and cluster properties, e.g., in their number and in component numbers.

In the case of autonomous entities, examples of local constraints in addition to the ones considered above include short-range rules, such as collision avoidance (separation rule) through some feedback mechanisms; adoption of a dependent behavior, e.g., imitation of that of the adjacent neighbors, and adaptive and self-regulatory behavior through some automatic mechanism.

### 3.2.2. Global Constraints

In the case of nonautonomous entities, examples of global constraints include environmental properties such as geometric properties, e.g., shapes and borders to be respected; the possible nonhomogeneous availability of energy; and the existence of possible influential and varying fields, such as the electromagnetic field. Other examples include consideration of the next change in criteria, such as probability, iterations, continuity, and optimization.

In the case of autonomous entities establishing collective behaviors, such as a flock, examples of global constraints in addition (we consider that nonautonomous entities are a subcase of autonomous entities: An autonomous entity may decide to behave as a nonautonomous entity, while the opposite is not true) to the ones considered above include, in 3D collective motion, the rules identified by Reynolds in [23]:

- Alignment rules: Individuals must control their motion so as to point towards the average motion direction of locally adjacent components;
- Cohesion rules: Individuals must control their motion so as to point towards the average position of locally adjacent components.

Other rules may represent anisotropic behaviors exhibiting properties with different values when measured in different directions, as in the case of anisotropic flocking, representing lateral weighing of the neighbors inspired by the experimental results.

Moreover, we may consider the random drift behavior producing a force which applies an acceleration to the entity, e.g., a boid, in a random direction. Finally, the homogeneous validity of gravity and the isotropic, invariant environmental conditions are considered.

Rather than metrical approaches, as described above, a topological approach is considered using topological constraints based on the phenomenological detection that starlings in huge flocks interact with their 6–7 closest neighbors [8,24–26].

Other examples of global constraints for autonomous entities in addition to the ones considered above include cognitive constraints, such as behaviors to be avoided for any reason; avoidance of scary situations; shared collective choices such as migration; and selections of averages in respective ranges of admitted values in the conceptual framework of implementing analogies.

A crucial question relates to how to understand microscopic characteristics of collective behaviors, e.g., long-range correlation and scale freeness, or mesoscopic, e.g., clusters or infraclusters, as rules to be respected that are suitable to "prescribe" the coherence of collective behaviors. We understand here that they should be more appropriately considered as properties, possibly generated by (also cognitive for autonomous entities) domains and constraints, as shown in Table 1.

**Table 1.** Constraints and domains.

| Constraints |
|---|
| (a)　The physical admissibility (e.g., achievability, admissibility, compatibility, continuity, probability, and subsequence) of changes that should respect the general physical evolutionary constraints of the phenomenon (such constraints will take the form of ranges of validity). |
| (b)　The compatibility intended as a local property between ranges of values of the neighborhood over time. |
| (c)　The equivalence of possible ranges of values available as inducing equivalent local and global effects (such equivalence is related to the unpredictability of processes of emergence when different options for the systems are equivalent and the selection may occur in different ways). |
| (d)　Local and global constraints (see Section 3.2). |
| **Local, Configuration, and Global Domains** |
| (e)　Local domains: Instantaneous domains, networks of constraints and degrees of freedom, and ranges of admitted values are supposed to equally iterate and be differently used over time by temporal sequences (different time granularities are possible) with different parameters assumed by single entities that are considered isolated. |

As outlined in Table 1, we consider the generic, local, and global constraints introduced above as degrees of freedom for entities constituting collective behaviors.

As introduced in Section 6.2, for each entity, we consider the conceptually dual domain of constraints and ranges of values possible for the subsequent change that is computable for nonautonomous entities.

More interestingly, for collective coherent entities, we consider the dual domain of configurations of constraints and ranges of values possible for the subsequent collective change that is computable for nonautonomous entities. Configurations of domains both represent and prescribe configurations of admissible, compatible, probable, coherent, and equivalent values from the current ones assumed by the entities of their collective behavior.

Instantaneous configurations of such values are the evolutionary, subsequent, authorized possibilities among which the entities may collectively select the one to be assumed.

See Box 2 for a summary.

**Box 2.** Summary No. 2.

---

*Generic Constraints*

Allowing physical admissibility and local compatibility.

*Local Constraints*

Such as limits, ranges of validity for short-range attractive and repulsive forces and energy exchanges; number of cluster properties and of components. Rules of collision avoidance (separation rule); analogy, imitation, dependence on contiguous behavior.

*Global Constraints*

Such as the keeping of correlated behaviors (coherence) through, for instance, alignment rules and cohesion rules.

---

## 4. The Classic Microscopic Understanding of Properties of Collective Behaviors

In this case, we consider populations of interacting entities, e.g., molecules, vehicles, and insects, acquiring coherence. The process of interaction is intended to take place when the properties and behaviors of one entity influence those of another.

The process of interacting occurs, for instance, though the exchange of energy, e.g., collisions and information in cases of autonomous entities.

Among processes which may occur within such populations of interacting entities, we mention here the occurrence of self-organization and emergence.

In the case of self-organization, the population acquires sequences of properties in a phase transition-like manner that have almost regular coherence, e.g., repetitiveness and synchronicities.

Examples include collective systems established by nonautonomous entities (see Section 3.1); the regular repetitiveness assumed by swarms of mosquitoes around a fixed light and behaviors of pelicans flying around stacks of trash (in both cases, there is an attractor); regularities, e.g., sinusoidal, such as the formation of queues in a traffic flow; synchronizations and remote synchronizations occurring when nonadjacent pairs of entities become substantially synchronized in spite of the absence of direct structural connections between them or intermediate mediating entities such as in the brain and networks [27,28]; and those belonging to the basin of an attractor.

In the case of emergence, the population of entities is intended to acquire sequences of properties (also self-organized) in coherent ways more than only regularly, e.g., correlated, following power laws and self-similarity. This is a generalization of self-organization. In cases of emergence, sequences are intended to dynamically acquire multiple different coherent properties.

Examples include collective systems established by autonomous entities (see Section 3.1) such as anthills; cities; flocks, shoals of fishes and swarms with multiple and changing shapes, densities, and directions; markets; telephone and transportation networks; and termite mounds.

The peculiarity is that, over time, they continuously remain coherent, allowing constant recognizability in spite of changes, e.g., parametrical and topological changes [29].

Coherence is understood, for instance, to be given by the presence of dynamically simultaneous properties, such as ergodicity, coherently variable synchronizations, remote synchronizations, (long-range) correlations, polarization and global ordering, network properties, power laws, belonging to the basin of an attractor, scale invariance (features do not change if scales are multiplied by a common factor), and self-similarity (similarity to a part of itself). The roles of the observer and scalarity are both crucial as cognitive frameworks, such as for constructivism and De Finetti's probability [30,31], which is impossible without the configurations of expectations and interest generated by the observer. This

corresponds to questions that are unavoidable in research and invention of experiments. This makes the phenomenological detection of the coherence of processes of emergence and its constant or variable persistence possible in the framework of theoretical incompleteness [9,32] and equivalent behaviors [5].

Populations and communities of collective interacting entities within which processes of emergence occur are said to establish collective behaviors.

We should consider that we may have spatial collective behaviors, more correctly termed collective coherent motions, such as flocks and swarms, and nonspatial collective behaviors, such as markets, coherent traffic signals, e.g., the Internet, and coherent financial transactions, where we can consider representations in geometrical nD rather than physical 3D space.

Here, we consider the behaviors and properties of collective behaviors. They characterize collective behaviors up to identifying autonomous (in that their properties and behaviors are not deducible by that of composing elements) entities termed collective entities or collective beings [6]. Properties of collective behaviors can be considered in the perspective of finding correspondence that can influence and foresee their emergence, evolutions, and tendencies (see Sections 9 and 10).

Examples of behaviors of collective behaviors mentioned above and possible types of classifications include their possible constant iteration, e.g., circularity or partial repetition; tendency to disintegrate and recompose; variability in the quantity of components; short- or long-term life spans; compliance with functional sequences, e.g., sinusoidal patterns representing variations such as density and average time of change; tendencies of different collective behaviors to merge or not; and (possible partial) oscillatory instability. Furthermore, we can consider particular behaviors assumed, for instance, by flocks in self-defense, such as escape from predators through strategies like inducing confusion or becoming unrecognizable (e.g., some shoals of fish), and food searching, as well as properties of sequences of collective sales or purchases (establishing, in this case, positive feedback).

Examples of properties of the behaviors of collective behaviors are topological as related, for instance, to shapes and holes; density and its variability; and geometrical contours and outlines. Other properties may relate to the resilience of the collective behavior in the face of external perturbations.

In this conceptual context, we may consider conditions, such as changes in properties, when a collective behavior starts a process of degeneration in noncoherent aggregation (leading to degeneration of the collective behavior established) and, conversely, when an incoherent aggregation starts a process of becoming coherent. Furthermore, we may consider processes of merging, i.e., when two or more collective behaviors assume the same coherence among their composing elements, becoming undistinguishable, and the splitting up of collective behaviors, e.g., assuming different coherences and eventually establishing spatial distances.

As we describe below, among the purposes of this article, there is the possibility to introduce the correspondence between global domains, properties, and behaviors of collective behaviors and the corresponding effective collective behaviors as a possible research subject, particularly when considering collective behaviors and their dual, self-generated, computed domains.

We conclude this section by mentioning how several approaches are available in the literature to model collective behaviors at the microscopic level, i.e., considering the mechanism of interaction among particles, of whatever nature they are. A very good overview is available in [8].

See Box 3 for a summary.

**Box 3.** Summary No. 3.

---

*Microscopic Collective Behaviours*

We consider populations of interacting entities, e.g., molecules, vehicles and insects, sellers and customers, and users of services. The process of interacting occurs, for instance, though the exchange of energy, e.g., collisions, information, or goods and money.

---

## 5. The Mesoscopic, Metastructural Understanding of Properties of Collective Behaviors

A further approach introduced allowing the less formal, tolerant approaches mentioned above consists of considering clusters of components rather than components constituting collective behaviors. Cluster and infracluster properties are supposed to be suitable to represent the collective behavior under study [33,34].

In this approach, the individuality of single composing agents is substituted by cluster(s) of belonging, which is suitable to take account of incompleteness and fuzziness. The advantage is that this approach generalizes and goes further the microscopic interaction-based mechanism that is inappropriate for collective behaviors because of different reasons, since interactions are assumed to

- Be possibly multiple;
- Be completely or partially superimposed per instant;
- Have variable intensity;
- Change over time; and
- Have different temporal durations and starting times.

Moreover, the microscopic approach is unsuitable when there is a high number (potentially infinite) of composing entities.

As introduced above, collective behaviors are self-organized and emergent, in that they do not follow predefined structures of interactions. The cluster-based approach is independent of the interaction structures and, as such, the approach is termed metastructural ([4], pp. 102–128).

Furthermore, the same composing agent may simultaneously belong to different clusters related to different variables, e.g., speed, altitude, and direction in spatial collective motion.

As is well known, clustering takes place as aggregations using suitable criteria, such as the similarities of values assumed by a variable, i.e., when its values fall within a validity range.

Different techniques of clustering are available. For instance, the statistical techniques of multivariate analysis [35] allow the selection of clustered elements by optimizing the differences of values. Furthermore, clusters among values may be computed as average values, centroids, or arithmetic means. This clustering is possible through the use of suitable and available computational techniques, such as k-means [36], where the objective is to minimize the total infra-cluster variance.

In real implementations, in order to allow comparability, the number of clusters should be constant along time. Further, it is possible to decide the number of clusters to be considered using appropriate computational approaches, such as the Elbow and Silhouette criteria (see, for instance, [37,38]).

We also mention fuzzified clustering, where each item can belong to more than one cluster [39].

Regarding the cluster properties, we mention how the ones considered for the microscopic case may also be used in this case, such as synchronization, correlation, network properties, power laws, and scale invariance. Furthermore, for each cluster along time, it is possible to consider the number of belonging elements and their distribution, for instance, in significant percentages close to the minimum, maximum, average, or random spread out, and the total time spent by each element to belong to a specific cluster.

Such properties are intended to be metastructural, i.e., suitable to generalize and summarize properties of intractable microscopic representations and identify, as well, globally significant collective properties that are microscopically irreducible.

The research issues are related to the mesoscopic coherence [33].

See Box 4 for a summary.

<div style="text-align:center">**Box 4.** Summary No. 4.</div>

---

*Mesoscopic Collective Behaviours*

　　Within populations of interacting entities, e.g., molecules, vehicles and insects, sellers and customers, and users of services, we consider instantaneous clusters of entities, i.e., grouped for significant similarities. Examples include the movement, speed, direction, acceleration, altitude, economic amount of the transactions, and services used in sequences or regularly.

　　Cluster and infra-cluster properties are supposed to suitably represent the collective behavior of the population under study when different, variable, and multiple interactions are microscopically intractable.

　　Examples of clusters properties include their synchronization and correlation in manifesting and their quantitative compositions over time.

---

## 6. Fields and Domains.

　　It is possible to consider other tentative different approaches than those considered in Sections 4 and 5.

　　We can consider alternative approaches to represent collective behaviors, allowing for explicative mechanisms different from the usual interactions-based mechanisms.

　　For instance, in Section 6.1, we mention the coherence of collective behaviors as intended in the framework of quantum field theory (QFT), as entanglement [40], and due to NG bosons.

　　However, constituting entities are intended here—as in classical physics for which the properties of quantum physics do not apply—to act in the classical understanding of collective behaviors established by generic diluted or disperse matter acquiring coherence.

　　On the basis of some concepts of physics, such as the original duality particle-wave, due to the intuition of Louis De Broglie (1892–1987); the unavoidable entanglement in quantum physics; and the particle as field, we tentatively introduce (Section 6.2 and below) conceptual possible approaches to consider processes of collective behavior with dual domains. We leave possible mathematical versions of these proposals for future research (see Section 10).

　　The idea is to consider a collective behavior as a configuration of modalities for the effective respect of generic, local, and global constraints, and equivalences which then become interdependent probabilities and are intended to establish domains.

　　The establishment of coherence through suitable selection mechanisms in domains should be considered suitable to replace the usual mechanism of interaction and its properties.

　　Interrelations, balancing, eligibility, compatibility, and compliance with suitable constraints starting from the initial conditions generate cascades of configurations necessary for guaranteeing coherence. In this conceptual framework, dynamics is intended as choices to break equivalent configurations.

　　Dynamics is also intended as an evolution of domains (structural dynamics in emergence as mutations of the interrelations contrasting with the use of constraints and ranges of validity of domains). Such configurations have their evolution given by the interdependence between the degrees of freedom and behavioral probabilities of the composing entities. We should also consider the behavioral properties of the domains themselves.

### 6.1. Fields

　　The consideration of fields may be suitable, for instance, when the number of interacting elements is very large, potentially tending to infinity, and when the individuation of single elements becomes difficult or even almost impossible due, for instance, to their equivalence, multiple roles, and variability.

　　As is well known in field theory, a physical quantity is assigned to every point in the space-time of the field.

　　The field is intended to be a physical–mathematical region of a space whose characteristics are functions of configurational variables, typically temporal or spatial. A field, like every physical system, has its own "configuration", and the variables describing this configuration are called configuration variables, e.g., geometric and kinematic variables describing the configuration of a system.

A vector field is an assignment of a vector to each point in space. Usual examples of vector fields represent the speed and direction of a moving entity or the gravitational force, changing from one point to another one in space–time.

Historically, fields were considered to be fields of forces, having values depending on space–temporal coordinates, suitable to describe the effects of forces produced by material sources.

The discovery of electromagnetism allowed the consideration of transmutations of a field into another kind of field. This is the case for the production of a magnetic field by the current generated by an electric field. Furthermore, in addition to the transmutations, there is the effect of propagations of fields, such as for the case of electromagnetic waves.

All this introduced the opportunity to consider the fields as primary entities for physics, rather than objects or sources which can be considered equivalent to specific space–temporal places characterized by very strong intense concentrations of a field.

In this case, we can use all the knowledge developed for the theory of fields.

In quantum field theory (QFT), see, for instance, [41], the supposed duality field-particle leads to the concept that there are no particles, there are only fields [42,43]. In quantum physics, particles are epiphenomena arising from fields.

We try to introduce the domains of coherence of collective behaviors by considering their unavoidable double nature or unavoidable environment in conceptual correspondence with the vacuum or vacuums in QFT [44].

In classical systems, an example of such an environment is given by ecosystems. In this case, " ... the environment pervades the elements which produce, in turn, an active environment. This environment, if we can still call it such, is active and not an amorphous, abstract space hosting processes. It is interesting to consider eventual conceptual correspondences with the quantum vacuum pervading everything." ([4], p. 13).

In QFT [45], the void is not emptiness or absence of everything but a pervasive, unavoidable source of properties, such as entanglement.

Furthermore, in the quantum vacuum lacking any particles, there are electromagnetic fields fluctuating about an expectation value of zero.

The quantum vacuum is intended to precede matter, and in such a way, it also must precede space and time [46].

Correspondingly, " ... emergence could be intended as coming first, as a property of pre-matter, of the vacuum. The quantum void could thus be intended as a kind of field of potentialities ready to collapse but always pervasive as are the probabilistic features of Quantum mechanics (QM)" ([4], p. 130).

Here, we tentatively try to conceptually consider this for collective unstructured entities (disperse matter like molecules, birds, and proteins or nonmaterial like information, signals, and money) such as collective behaviors for which a-structured approaches were already introduced in Section 5. We tentatively try to introduce a further approach of considering the dual nature of collective behaviors as collective entities and their domains of real possible states.

In particular, we consider the coupled collective behavior–dual domain, which is hypothesized to have reciprocal influence, and the domains, which are assumed to have autonomous evolutionary properties. This duality occurs as in physics, the wave-particle or field-particle can be considered according to convenience as two different nonequivalent levels of description.

Coherence and NG Bosons

We mention that, in QFT, thanks to the entanglement, no classical interactions are required to make entities interdependent. This relates to the unavoidable pervasiveness of the quantum vacuum or vacuums given by the variety of possible states of vacuum ([4], pp. 83–87).

"We will try to tentatively consider structural properties of the hosting space as source or generator of systemicity (the conceptual inspiration is given by quantum entanglement) when a less totalizing

and invasive view is given by properties of the non-separable environment. We will consider the eventual propagation of systemicity." ([4], pp. 171–172). In QFT, the void is considered as a pervasive, unavoidable source of properties such as entanglement, making systemicity an unavoidable, intrinsic, embedded property.

For instance, in the quantum vacuum, each perturbation causes the emergence of collective long-range excitations named Nambu–Goldstone bosons (NGBs), which coordinate the behavior of individual components of the system, so as to keep general coherence.

Moreover, the NGBs can interact among them, giving rise to the appearance of macroscopic entities (the so-called quantum objects), which, in turn, modify the behavior of the entire system from which they originated.

Thus, NGBs act as "coherence keepers", a role characterizing one of most important aspects of emergence. On the other hand, this emergence is far from being unpredictable, being determined by a specific choice made by the external environment ([4], pp. 243–247).

Moreover, "... the plasticity itself of QFT conceptual structures does not preclude the unexpected occurrence of new models and new achievements helping to better understand what is emergence" ([4], p. 248), [47].

They are the basis for a quantum-systemics, implicit in the quantum description of the world. See Box 5 for a summary.

**Box 5.** Summary No. 5.

> *Fields*
>
> A field is intended as a physical–mathematical region of a space whose characteristics are functions of configurational variables, typically temporal or spatial.
> As is well known in field theory, a physical quantity is assigned to every point in the space–time of the field. Well-known examples include the gravitational and electromagnetic fields. The theoretical interest is given by the fact that a field generalizes a physical property in the space.

*6.2. Domains*

The concept of domain considered here is different from the one used in mathematics such as, for instance, the domain of functions in information technology (IT) when dealing with web domains and magnetic domains in physics, which are intended to represent regions of a magnetic material in which magnetization is in a uniform direction. The concepts of domain and region are considered here to be equivalent. However, we use the term domain.

Felds and domains contrast in that in a field, each point is supposed to have a precise value of a definitory variable, e.g., gravitational and electromagnetic, while the domains are intended as regions of possible options of the space of degrees of freedom virtually generated by an entity. The domain corresponds to multiple options representing possible admissible states for the entity that respect both the constraints and degrees of freedom (dimensions of the space). In sum, a domain is assumed to represent, through (configurations of) ranges of values, the subsequent changes possible for the (collective) entity. Entities may be material, e.g., mechanical components, or nonmaterial, e.g., shares, with effects on the nature of constraints and domains.

While, in a field, only one value is available in each point for the field variable, for domains, multiple achievable, admissible, subsequent, compatible, incomplete (partial or fuzzy), equivalent, and nonequivalent choices are available [5], and only one (in nonclassic worlds, more contemporary states are possible) should be chosen. The domain is assumed to be represented over time by multiple authorized intervals of values available for the subsequent change. The domain is intrinsically time-dependent, as each domain ($t_{n+1}$) depends on (multiple) previous domain(s) ($t_n$). We use the expression "*domain*" to represent *domain(t)*.

The freedom for the changing possible to an entity is represented by the characteristics of its domain. The domain represents the potential dynamics, transformation.

See Box 6 for a summary.

---

*Domains*

The domain of a single entity, considered as isolated, is constituted by all the options, the changes subsequently *possible*. Examples are physical and chemical, *achievable* choices. In general, no *heavy* (the reference is to scalarity) discontinuities are allowed; e.g., in classic physics, an entity cannot appear and disappear or simultaneously reappear in different locations. Furthermore, *admissible* choices depending on previous values, for instance, an entity cannot have two very different speeds at two subsequent instances and not be in a location very distant from the previous one. *Admissibility specifies achievability*.

---

6.2.1. A Basic Conceptual Understanding

In order to specify the concept of domain of a single entity, e.g., a material or a process, and of collective entities, e.g., markets or flocks of boids, we first consider all of the concepts of degrees of freedom available to the entity. In short, the degrees of freedom are the independent variables (completely or significantly) representing the possible behavior and the changing of the entity under study. When considering a punctiform vehicle, elementary examples of degrees of freedom are its speed, direction, and acceleration (neglecting, for instance, friction, fuel consumption, vibrations, instabilities, and road holding, which become predominant when at high intensity and are represented by changes in the values of the variables of the degrees of freedom).

The degrees of freedom are assumed to specify the space of variables in which a point represents the state of the entity over time.

Each variable, i.e., dimension of the space of degrees of constraints, has constraints to be respected by the entity.

In elementary cases, such constraints take the form of the min–max range of values for each variable, i.e., the degree of freedom that is not overcome.

In more realistic cases, the ranges of values for each single variable may be multiple and disconnected.

In elementary cases, the ranges of values representing constraints are assumed to be constant, given by properties of the entity independently from the context. The entity is assumed to always be subject to the same behavioral limitations independently from:

- previous choices; and
- the choices of other entities.

As we said above, variables are intended to be independent, and so their ranges of values specify constraints.

The subjects related to ranges of values specifying constraints may be considered in ways different from the classical approaches where once they have been defined, there is no more to add.

For instance, in [4], we considered that it is possible to take into account what happens (at suitable levels of scalarity) between degrees of freedom.

The mesoscopic area between is the place where emergent phenomena occur [4].

Moreover, when dealing with constraints, e.g., min and max, in a simplified understanding, it is sufficient that they are respected.

In a more sophisticated way, we considered how they are effectively used, spent over time, e.g., for the majority of the time close to the max, or to the min, or to the average, or in random ways.

As an elementary example, we may consider the domain of an entity, e.g., a boid, in a 3D isotropic, uniform space.

In this case, domains are "local" regions, i.e., related to a single, supposedly isolated entity, constituted of degrees of freedom with constraints, subsequent in a discretized time.

The domain of a single, isolated entity at time $t_n$, considered at a suitable scalarity and time granularity, may be intended to be given by a set of fuzzy, i.e., tolerant small variations and imprecisions (for instance, of min and max values) of values of degrees of freedom, with their constraints and starting from previous values, for instance;

- Achievable choices: We consider changes that are physically achievable from the current status in the context of invariable properties as in rational mechanics and classical physics defined by *laws* assumed to be objective and unchangeable. In general, no heavy (the reference is to scalarity) discontinuities are allowed, e.g., in classic physics, an entity cannot appear and disappear or simultaneously reappear in different locations;

- Admissible choices: Depending on previous values, for instance, the speed, direction, momentum, and altitude at time $t_n$, we consider fuzzy ranges of values that are dynamically admissible for time $t_{n+1}$. Furthermore, achievable choices may be not admissible starting from particular values at time $t_n$, e.g., an entity cannot have two very different speeds at two subsequent instances and not be in a location very distant from the previous one: admissibility specifies achievability;

- Subsequence of choices: Subsequencing in classical physics should be considered at suitable scalarity and time granularity levels, avoiding events occurring between choices invalidating general achievability and admissibility;

- Equivalent and nonequivalent choices: Equivalence and nonequivalence are related to the properties of the dynamics. For instance, a choice may be equivalent for the dynamics of a single entity, since maintaining properties is intended to be crucial for the identity of the dynamics, e.g., positions on a table are equivalent, except those at the edges which foreshadow a loss of stability. We may consider how some different changes in collective systems may, however, not mutate the straightness or circular fly of an entity such as a flock or initiate disaggregation: admissible choices may be equivalent or not;

- Compatible choices: We refer to the compatibility of choices of infra-domains, since the *independence of variables* representing degrees of freedom should be limited with respect to some phenomenological interdependence. For instance, for boids of a flock, significant changes in internal positions cannot occur with no or insignificant changes in speed. Levels of interrelated degrees of freedom allow compatibility;

- Probable choices: We refer to ranges of probabilities depending on previous temporal evolutive stories that are context-dependent when, however, low probable choices may occur anyhow because of external environmental perturbations;

- Incomplete choices: We refer to choices intended to occur when the process of change is incompletely performed, for instance, suspended, such as in cases when the initial energy is, for some reason, only partially spent, suspended or inconstant. Furthermore, the adopted change may be partial compared to that of neighbors, or when compatibility is partial and should be restored with regard to some significant number of degrees of freedom. This may allow temporal, instantaneous breaking of rules (in case of radical emergence, see below). Such events may be recovered by subsequent processes resuming properties and coherence (see Sections 7.2 and 8.1) or may give rise to new regimes of degrees of freedom establishing new admissibility, compatibilities, and equivalences.

All of the aspects listed above identify, for each entity and per instant, the current possible choices as options for the selection of the next changes to be assumed (the option of no-change should also be considered). The choices relate to intervals or ranges of values for constraints of the degrees of freedom available.

See Box 7 for a summary.

<div style="text-align:center;">

**Box 7.** Summary No. 7.

</div>

---

<div style="text-align:center;">

*Domains of Collective Entities*

</div>

We consider the domain of collective entities and not of their composing entities. The domain of collective entities is not the sum or superimposition of the individual entities. Options and changes available in the domain relate to the comprehensive collective systems, such as topological; related to internal distribution, density; index of correlation, ergodicity; and average speed, direction, altitude, and amounts (economical, of information). The averages may be substituted by more sophisticated statistical measures. The domain corresponds to multiple options representing possible admissible states for the entity that respect both the constraints and degrees of freedom. In sum, a domain is assumed to represent, through (configurations of) ranges of values, the subsequent changes possible for the (collective) entity.

---

The crucial conceptual point, as elaborated in the following text (see Sections 7 and 8), is how the dynamics and coherence become matters of proper *selection* among the available choices (see Sections 7.2, 8.1 and 10). Table 1 summarizes the concepts introduced above.

6.2.2. The Domains in More Detail

Now, we present some more specific aspects of the concept of domain under consideration here. At first, we consider how domains can be effectively identified.

Dealing with collective beings, domains are, in reality, cognitive spaces for use by entities provided with cognitive systems. The behavior within cognitive spaces or cognitive domains is not computed but, rather, induced, perceived, analogically decided, remembered, imitated, attempted, and evoked ([6], pp. 104–113). Cognitive domains may, however, in this case, be represented by some *learning* computations, e.g., machine learning and deep learning (see Section 10), which are suitable to use and generalize individual behaviors over time. Individual behaviors may be detected using suitable techniques such as stereometric and computer vision techniques used, for instance, to measure 3D individual birds' positions in compact flocks, for example, of up to 2600 European Starlings (*Sturnus vulgaris*) in [24], and based on data of collective behaviors provided by global positioning systems (GPS). Other cases apply to self-tracking entities, such as financial events and the collective behavior of stock price movements [48].

We mention two approaches to consider effective domains of nonautonomous entities.

The information available in Table 1, as ranges of values allowing for changes in entities, may be inserted by the researcher as abstract constraints that are technologically supported, extrapolated, and learnt from a significative number of subsequent GPS-detected steps of the collective behavior under study with inductions allowed. The domains and files of Table 1 are interesting as they allow the profiling of collective behaviors.

The second is based on considering data made available by signals generated by suitable electronic devices [49] and by simulations, rather than by the GPS, of collective behaviors and information of the instantaneous statuses of collective entities. This is, for instance, the case for the flock simulator available at [50] where both Cartesian ($x$, $y$, $z$) and spherical ($r$, $\theta$, $\phi$) coordinates are available.

As introduced above, here, we consider domains of the authorized possibilities "around" an entity, e.g., usual macroscopic objects, particles, and waves [51] and nonmaterial entities. The achievability and its possibility are expressed by the constraints considered above as the physical admissibility and compatibility. Their equivalences (for any number) establish the domain of possible changes induced by the existence and properties of single entities. The dynamics represented in such virtual space should be suitably studied.

At this point, we specify that the authorized possibilities are not reduced to dynamic changes of the material entities in time and space but also relate to possible structural changes when the entity structurally mutates, e.g., morphogenesis and after a transition phase when the entity is no longer the same, and, more generally, when structural interactions in collective coherent entities change in duration, intensity, and types.

On the other hand, the same domain also identifies the dominium of the nonachievable possibilities as possibilities that do not belong to the domain.

However, the belonging considered above may be fuzzy, allowing the establishment of fuzzy domains and quasidomains, as introduced below.

Furthermore, the domain and the role, as the use of the domain, performed by an entity can be considered as a specific materialization or realization, due to any reasons, of phenomenological properties, e.g., energetic for nonautonomous entities and cognitive for autonomous entities, and related to a previous, pre-existing (see Section 6.2.3) domain of authorized possibilities. The first aspect connects the domain with phenomenological properties and does not just model them or represent acquired properties.

At this point, it is more interesting to consider the domain of authorized possibilities related to a collective population of dynamic entities, rather than one related to single entities only. In the second case (single), we deal with fixed structures among entities, more precisely, relations. In the first case (populations of collective dynamic entities), we deal with structural dynamics when interactions among entities are changing from the previous instance and have different durations, different intensities—interfering and superimposing—and are differently multiples over time. Interesting cases arise when collective populations establish coherences, such as for collective behaviors. In this case, the authorized possibilities of the collectively behaving population, i.e., assuming continuous or quasicontinuous coherence—the ability to recover and tolerate temporary levels of incoherence (see Sections 7.2 and 8.2)—relates to the entire population, are irreducible to sums or sets of authorized possibilities related to single composing entities. Quasicontinuous coherence, the ability to recover and tolerate temporary levels of incoherence should be considered when dealing with real cases with a variable number of mutating components and radical emergence.

Local domains are intended to represent the degrees of freedom available—the states that can be assumed by single separated, isolated entities.

While for single, separated and isolated entity domains, the limited choices border on determinism mitigated by equivalences and probabilities, the situation is different for collective populations of interrelated entities such as with collective behaviors and interrelated, networked configuration domains, i.e., clusters of local domains.

Global, interrelated, and in some cases, networked, configuration domains can be considered as having some autonomous, acquired properties, i.e., different from the ones of single local domains and configuration domains. Examples are network properties where nodes are configuration domains. Network properties include scale freeness (when the network has a high number of nodes with few links or a small number of nodes with a high number of links. The scale freeness of a network correlates with its robustness to failure by establishing fault-tolerant behaviors; the establishment of small worlds (when non-close neighbor nodes can be reached from every other node via a small number of intermediate links); the cluster coefficients (intended as a measure of the likelihood that any two nodes possessing a common neighbor are themselves connected); and the degree distribution (while the degree of a node is intended as the number of neighbors, the degree distribution is the probability distribution of the node degree over the entire network). Examples of properties of configuration domains include topological domains related to stability, continuity, and oscillatory factors. Furthermore, we can consider the evolutionary dynamics of global configuration domains as being representative of the phenomenological dynamics of the corresponding represented collective behavior. As we discuss hereafter, this opens the way for more abstract (properties of domains) and, at the same time, more practical (domains extrapolated from real behaviors) representations of collective behaviors and considers properties of interrelated configuration domains suitable for indirectly and nonlinearly inducing changes in the collective behaviors themselves.

The domains of possibilities related to a population of collectively behaving entities are then given by authorized possible configuration domains—in some cases, just equivalent (same coherence), and in other cases, nonequivalent (different coherence or temporary partial coherence or incoherence).

However, the last case may initiate new evolutionary paths specifying subsequent nonequivalent unique behaviors to the collective coherent population. This is the case, for instance, for a flock with circular behavior switching, for any reason, to a straight flight or assuming a self-defense behavior in the face of a predator.

The global configuration domains, networks of possible configuration domains for the entire coherent collective behavior, dual of the collective behavior, could, however, be effectively computed from the instantaneous configuration of the phenomenological collective behavior. However, the computation of such global configuration domain networks is expected to be analytically intractable if not for simplified cases.

Accordingly, we may speak of the self-generated global domain.

The existence of the global domain may be, at first, considered as an abstraction having properties suitable to generate or interact with other domains.

The thesis we have in mind is that any collective behavior brings together its self-generated general domain and that such domain acquires autonomous properties.

See Box 8 for a summary.

**Box 8.** Summary No. 8.

---

*Experimental Detection of Domains of Collective Entities*

*The Case of Collective Nonautonomous Entities*

Domains may be computed in the case of self-tracking entities such as financial events and the collective behavior of stock price movements. Furthermore, the information available in Table 1 as ranges of values allowing for changes in entities may be inserted by the researcher as extrapolated, computed from a significant number of subsequent cases, and historically GPS-detected steps of the collective behaviours under study. Such domains allow the profiling of collective behaviors. Another possibility is based on considering data made available by signals generated by suitable electronic devices and by software simulations.

*The Case of Collective Autonomous Entities*

Historical, phenomenological individual behaviors may be detected using suitable techniques such as GPS. Another possibility is the stereometric and computer vision techniques used, for instance, to measure in 3D individual birds' positions in compact flocks. The sequences allow computation of the domain and fix phenomenological types of coherence.

Dealing with collective beings, i.e., entities with sophisticated cognitive systems, such as human beings, domains are, in reality, cognitive spaces for use by the entities. The behavior within cognitive spaces or cognitive domains is not computed but, rather, induced, perceived, analogically decided, remembered, imitated, attempted, and evoked. Cognitive domains may, in this case, however, be represented by some learning computations, e.g., machine learning and deep learning, which are suitable to generalize individual behaviors over time.

---

### 6.2.3. Pre-Existence of the Domains

As for the eventual pre-existence domains, which, hypothetically, may have no corresponding generative material or "real" components, we may consider the process of establishing initial conditions for the possible pre-collective behavior of entities requiring unruled tentative interactions among early potential components. The first simplified interactions may be assumed to be initial conditions and to respect only basic requirements such as the request to not collide while behaving in admissible and mutually comparable ways, i.e., respecting min–max ranges and modalities as behaviors mainly close to specific values or changing regularly or moderately (with regard to scalarity) at random. It should be intended to be a self-establishing convergent process. Examples include spontaneous synchronizations (applauses, objects on vibrating surfaces, fireflies).

However, phenomenological initial conditions may be more precisely ruled by more specific requirements of admissibility and compatibility that are gradually set through confirmatory and try and try again iterations.

Further, the domain should be considered as the pre-existing reality encountered by any new aspirant entering entity, forcing it to adapt when trying to belong and become a member of a hosting collective behaving population, e.g., when entering a vehicle traffic line.

Furthermore, the structural dynamics of the collective behaving population make the belonging of all the autonomous entities new in each instant, so they are requested to continuously restore and maintain coherence in the framework of different parametrical subsequent values.

However, we specify that domains and their pre-existence cannot be reductively intended as kinds of Laplace machines, supporting the idea of a deterministic world even if with fuzzy constraints. Equivalences of possibilities, equivalent configurations, and probabilities are solved when the collective beings make choices and collapse through fluctuations or cognitive collective decisions.

The domain of possibilities related to a population of collectively behaving entities is then operative during its behavior, rather than understandable as due to the consequence of its existence. As for the case of a single entity, we may consider the same domain as the identifier of the complementary domain of the non-achievable possibilities (nonallowed possibilities). However, such belonging may be fuzzy, allowing the establishing of equivalences and fuzziness crucial for the processes of emergence based on quasicoherence.

See Box 9 for a summary.

**Box 9.** Summary No. 9.

---

*Pre-Existence of Domains*

Domains may be intended as pre-existing the corresponding collective behaviour when initial conditions constitute a self-establishing convergent process. Examples include spontaneous synchronizations (applause, objects on vibrating surfaces, fireflies). Initial conditions tend to become predominant upon random changes, and for entities becoming involved for some reason.

---

## 7. The Case of Domains of Collective Entities

As mentioned above, a completely new scenario opens up when considering a collective interdependent population of entities with their own individual domains, i.e., ranges of values for the constraints of their degrees of freedom. Because of the interrelations among the entities, domains are superimposing. Superimpositions may be, however, empty. In this case, individual, independent changes are allowed.

One single domain is supposed to have several superimpositions with diffusive effects due to choices performed within one of the superimposed domains. We consider configuration domains and global domains.

### 7.1. The Case of Domains of Roles in Collective Systems

As considered above (see Section 3.1), we distinguish among collective systems established by nonautonomous entities and those established by autonomous systems.

We consider local, configuration, and global domains of authorized changes as degrees of freedom.

Now, we consider domains as being constituted by roles available to be chosen and performed by entities.

A role for a generic entity is intended to have, at time $t_n$, specific values among the ones authorized by the domains, for instance, for acceleration, direction, distance, and a defined temporal duration within the degrees of freedom available. A role is a usage of the domain represented as a vector of specific values or ranges of possible values, among the ones authorized by the domains. For future research, we may consider the case of vector roles of variable, numerable dimensions.

In cases of nonautonomous entities, roles can be understood as energetic effective resultants (summations in linear cases) due to exchanges of energy, variations, and inhomogeneities in the availability of energy, environmental constraints, material properties, and wearing out. Due to the

probabilistic properties of resultants, roles are anyway usually consequent (until they are emergent), nondeterministic, and even compatible with catastrophes.

In case of autonomous entities, roles can be understood as being due to decisions made through cognitive systems of different levels of complexity.

Dealing with collective behaviors of collective beings, we introduce the concept of "In emergence processes . . . the agents can take on the same roles at different times, and different roles at the same time", by "assuming the conceptual interchangeability of agents, playing the same roles at different times" ([6], pp. xiii, 107). This is well represented by collective interacting populations of entities assuming an ergodic behavior ([6], pp. 302–305).

Dealing with collective beings by considering spaces of domains as available roles for entities opens up questions such as those related to the selection mechanisms (see Sections 7.2 and 8.1) of domains for entities allowed for multiple choices and how selection mechanisms in domains allow coherence (see Section 8).

Roles are intended to be equivalent when the "same" emergent properties are maintained by the collective being of belonging.

## 7.2. The Coherence of Domains of Roles for Collective Beings

At this point, we elaborate on the approach by considering that roles are not computed by autonomous entities by applying some rules, e.g., of interaction and constraints, but, rather, are selected among options such as those listed in Section 6.2.1 as being available, compatible, continuous, possible, admissible, distinguishing between equivalent and nonequivalent, and probable.

Composing entities of collective beings is a matter of conceptual cognitive selection of roles where the process of selection should be not reduced to processes such as optimization, but, rather, should be considered as the constructivist continuous cognitive generation of options in accordance with the properties listed above.

As we said above, for autonomous entities, coherence should not be intended as a constraint to be respected, but, rather, in processes of emergence, it should be considered as a property to be continuously, approximately built in real time. It is a matter of quasicoherence that is irregularly, nonhomogeneously, and continuously acquired at different levels until it resumes from extreme minimums and changes in types (structural changes). Violations of constraints in the case of autonomous entities may occur because of a variety of reasons, including environmental turbulences and their restoration after a drift. Quasicoherence is intended to not always be a coherence and not always be the same coherence.

The generative mechanism of coherence may be intended as a sequence of cognitive choices of representable, explicable, and modellable roles, as performed by usages of approximative rules and constraints of the domains. In simplified contexts, more realistic quasicoherence (incomplete coherence) degenerates in (complete) coherence, as intended by the generated rules [23].

The infraconfiguration transient coherence relates to phenomena of quasicoherence (see the concept of quasisystems in [4] and theoretical incompleteness in [9] when coherence may involve variable percentages of entities). The percentage and duration of sequences of quasicoherence should be acceptable, predominant, tolerable, and recoverable (depending on scalarity) to maintain collective behaviors.

Tolerance should be measured, for instance, in terms of percentages, variations, and periodicity, allowing interesting profiling information.

The detectability (recognized as such) of significant percentages of some infraconfiguration regularities shown by quasicoherence allows identities to be given to collective beings.

See Box 10 for a summary.

**Box 10.** Summary No. 10.

---

*The Case of Domains of Roles in Collective Systems*

A role for a generic entity is intended as simultaneous assumptions over time of different options among the ones authorized by the domains. A role should be intended as a vector of specific options, values or ranges of possible values among the ones authorized by the domains.

Domains of roles can be intended as domains of clustered available options.

Examples of roles are the simultaneous assumptions of values, among the ones authorized by the domains, for different variables such as energetic, positional, and dynamical.

The coherence, i.e., the selection of coherent roles, is mainly energetically replicated, as for molecules in whirlpools, chemical reactions, and intended as self-establishing convergent process such as spontaneous synchronizations (applause, objects on vibrating surfaces, fireflies).

*The Coherence of Domains of Roles for Collective Beings*

Dealing with collective beings, e.g., flocks and swarms, domains are, in reality, cognitive spaces for use by entities provided with cognitive systems. The behavior within cognitive spaces or cognitive domains is not computed but, rather:

- induced, perceived, analogically decided, remembered, imitated, attempted, evoked;

  and

- selected among options of roles as being available, compatible, continuous, possible, admissible, distinguishing between equivalent and nonequivalent, and probable.

The coherence, i.e., the selection of coherent roles, is mainly cognitive, having consequent analytical, computational aspects such as synchronization, (long-range) correlations, self-similarity, and networking. In this regard, we underline how collective beings are monospecies sharing the same cognitive system.

---

## 8. The Generated Coherence

Here, we distinguish between properties of coherence and phenomenological properties, represented by generative mechanisms that are assumed to be responsible for the establishment of coherence.

Properties such as long-range correlations, ergodicity, power laws, and self-similarity are interdisciplinary and pervasive and are suitable for detecting and modeling coherence in phenomena such as in biology, physics, and economics.

Properties may play the role of generative mechanisms for simulations.

Mechanisms are constituted of rules in simulations and models, e.g., [23].

Real phenomenological generative mechanisms for both autonomous and nonautonomous entities should be considered as peculiar for a very large variety of cases, and generalizations are expected to only be parametrical and suitable for each typology.

We consider how such a large variety may be tractable by conserving and not neglecting significant peculiarities when using domains and selection mechanisms among roles.

### 8.1. The Generated Coherence for Collective Beings

Phenomenological mechanisms are specific for autonomous entities.

In collective systems established by nonautonomous entities, the coherence of roles can be understood as being due to environmental constraints, iterations of initial conditions converging to optimization through energetic selections, constitutive material properties, and conservative aggregations.

In collective beings, the coherence of roles can be understood as being due to the properties of the specific cognitive system possessed by the living entities and the consequent selection cognitive mechanism. Such properties may be combined with those considered for nonautonomous entities, which are non-predominant in this case.

The main aspect of the coherences of collective beings is their specificity for species, e.g., flocking for birds, swarming for insects, banks for fishes, and herds for mammals. Furthermore, there are specificities within such categories such as birds flying in V-shaped flocks and ducks and wild geese having a leader. The underlying mechanism may be intended to be the aggregative mechanism in motion (see, for instance, [52,53]).

In this regard, so-called naturally-inspired computations have been introduced, such as evolutionary computing; the genetic algorithm; genetic programming; particle swarm algorithms; ant algorithms; bacterial foraging algorithms; social algorithms; neuroevolution algorithms; artificial immune system algorithms; developmental and grammatical computing; grammar-based genetic programming; grammatical evolution; grammar and genetic programming; genetic regulatory networks; quantum-inspired evolutionary algorithms; plant-inspired algorithms; and chemically inspired algorithms [54,55].

However, phenomenological coherence remains a matter of research for several contexts, as "Whatever the origin of the scale-free behaviour is, . . . , the fact that the correlation is almost not decaying with the distance, is by far the most surprising and exotic feature of bird flocks. How starlings achieve such a strong correlation remains a mystery to us." [56].

Other cases take place for other species, such as in animals having geometrical accuracy in building nests such as space-occupation optimizer hexagonal cells in beehives, bird nests, colonies and the geometrical nongenerative mechanism properties of spider webs.

Further, other properties, such as self-similarity and synchronization, are a pervasive occurrence in natural, vegetal (fractals as leaves and broccoli), and animal (fractals as snail and clam shells) systems.

We may assume a variable balance between cognitive and noncognitive roles.

Options guaranteeing such properties, for instance, moves in space, may be supposed to be perceived, considered, and selected, by the cognitive system of the autonomous entity through processes of natural computation.

The crucial point is that such properties are detected ex-post as mathematical and computable while they are generated by peculiar phenomenological mechanisms. Such properties are not computationally acquired.

At this point, we may realize the nonreducibility of the phenomenological coherence of collective beings to their property. Furthermore, the acquired properties of collective beings such as aggregation and disintegration, altitude (in case of flocks and swarms), behavior, coherence, density, direction, shape, and speed may be changed by suitable actions on the generative mechanisms, e.g., energy-based, environmental, and perturbative.

The subject is of interest when considering collective systems of nonautonomous entities to be, for instance, disactivated, such as hurricanes (whirlpools), or defended, such as collective plantations attacked by a parasite (olive tree plantations attacked by the Xylella bacterium). Furthermore, the subject is of interest when considering collective beings whose properties should be properly maintained under increasing fluctuations or degenerations having any origin, and suitably varied, such as human social systems, for instance, economical, cities, schools, hospitals, and companies (e.g., increasing the behavioral quality); electorates (e.g., increasing the turnout rather than to manipulate); markets (e.g., orienting consumption in an ecological sense); and users of services (e.g., increasing self-learning abilities).

### 8.2. Repaired, Restored Coherence

We can consider realistic and admissible cases when the phenomenological selection mechanism fails to determine coherent choices, allowing temporary incoherencies. This failure may be admissible and occur for any realistic reason, such as mistakes and out of time choices, particularly for autonomous entities.

Furthermore, such a situation compels us to distinguish between the selection of coherent roles and the selection of changes making other previous, supposedly possible, incoherent changes to

become coherent, making the incoherent changes coherent through subsequent suitable recovery of the selection of roles. In this second case, the coherence is intended to be repaired and restored. It is the changes in the context that make local or previous incoherence become coherent. This may be done through a variety of equivalent changes. Temporary incoherent changes are supposed to be necessary to maintain the nature of emergence, over-riding iterative, unrealistic, constant coherences, such as self-organization. Such incoherence may be intended to be occasional, irregular and nonhomogeneous with respect to constraints and incomplete application of rules. Such events may be at the basis of processes of radical emergence. See the concept of quasicoherence introduced above in Section 7.2.

See Box 11 for a summary.

**Box 11.** Summary No. 11.

---

*The Proposed Paradigm-Shift*

The conceptual paradigm-shift contained in the text consists in tentatively replacing the interaction-based mechanisms used to understand and model collective behaviors with approaches based on domains and selection mechanisms among roles suitable to generate and maintain coherence.

*Quasi-Coherence*

The reason to introduce such a paradigm-shift is to allow no-idealist approaches suitable to deal with the more realist quasicoherence: When the coherence does not constantly apply to all the composing entities, different forms of coherence apply. However, the percentage and duration of sequences of quasicoherence should be acceptable, predominant, tolerable, and recoverable (depending on scalarity) to maintain collective behaviours. Tolerance should be measured, for instance, in terms of percentages, variations and periodicity, allowing interesting profiling information.

---

## 9. Final Remarks

As mentioned above, it is possible to consider not only admissible and compatible variations, e.g., by reducing available options, combinations and perturbations, e.g., by forcing variations, but also forms of propagations, transmutations or mutations of domains.

Moreover, research issues may relate, for instance, to the possibility of categorizing; identifying constants and specific, incomplete, quasimechanisms with predominant, variable characteristics; and figuring out the characterization of evolutionary properties of the domains and paths shown in Table 1, such as topological, continuity, stability, and oscillatory paths (see Section 6.2.2).

In this latter case, we can set possible evolutionary rules: For some time-intervals, we may ask what should be considered to come first: the global domain and its evolution or the collective behavior and its evolution? We underline the conceptual differences of considering anticipation by considering domains and by considering models. This is of interest for simulations.

Such an understanding of the mutual relationship between collective behavior and global domains is not abstractly linear or perfectly homogeneous, but, rather, it is dynamically dependent on the nature of the behavior of the elements and is composed of different and varying values and constraints. It should be intended to be a proper, continuous balance between multiple different and varying local and global constraints, and context-dependent ranges of values. The balance also concerns multiple microscopic, mesoscopic, and domain-based approaches (see Section 10).

There is a variety of ways to be coherent, such as the emergence of life which may be maintained and supported in a large variety of ways through structural changes due, for instance, to aging, diseases, and healing.

In sum, we considered the possibility of regarding collective behaviors as coupled domains of instantaneous validity of configurations respecting constraints and ranges of validity for variables, together with a selection mechanism (more are possible).

This understanding may conceptually replace the classical microscopic, interaction-based modeling approach and extend and generalize mesoscopic-based modeling by replacing clusters and infraclusters of entity properties (see Section 5) with infradomain properties.

Moreover, we should consider that properties of domain configurations may influence phenomenological collective behavior, allowing the emergence of new properties, e.g., topological, density, regularities in variability, shapes, statistical, ergodic, and behaviors of collective behaviors, e.g., diffusion, iteration, and resilience.

We may intend for the system of entities to explore, perform, and define the domain configurations, allowing collective behaviors and properties in the conceptual framework of quasicoherence.

Further, the domain may expand in terms of the number of elements and spatiotemporal validity, having effects on and incorporating entities while getting in touch with their domains.

Furthermore, domains may give rise to other domains. For instance, other domains may be induced through the adoption of analogical, influenced behavior within noncoherent, disperse, getting-in-touch collective entities. Cases occur when the same constituting elements are subject to multiple different domains (compatible up to certain levels) or a disperse independent element, with its local domain, which begins to belong to the general domain after interactions.

More interdependent, collective domains may give rise to a general resulting domain with new (overall, emergent?) properties.

What are the advantages to considering the domains of collective behaviors?

A mesoscopic approach has been introduced based on infracluster properties that is suitable for summarizing intractable microscopic descriptions and properties at a higher level.

The advantages of considering domains include the possibility of avoiding an intractable high number of elements (potential infinity) by considering any combinations, variations, and extensions of properties of the domains (the interest is in the number of equivalent configurations available); the identification of characterizing evolutionary rules; outlining behaviors of collective behaviors; and setting the properties of the domain as characteristics of identity of the corresponding collective behavior together with the selection mechanism. See Box 12 for a summary.

Furthermore, behaviors and properties of collective behaviors may be suitably expressed when considering a domain's properties, for instance:

- The maximum number of equivalent configurations available per instant;
- The number of equivalences (i.e., how many equivalent configurations are available per instant) and their variability over time;
- The usage of equivalent configurations (how many times they are used);
- Clusters of configurations, their properties, and infracluster properties allowing a kind of metastructural analysis on such clusters;
- The number of possible local communities of configurations of available changes;
- Topological distances among configurations;
- The number of entities subject to single configurations;
- The number of entities simultaneously belonging to different equivalent or nonequivalent configurations

**Box 12.** Summary No. 12.

---

*Final Remarks*

The conceptual approach introduced considers collective behaviors as coupled domains of instantaneous validity of configurations respecting constraints and ranges of validity for variables, together with a selection mechanism (more are possible).

Furthermore, domains and selection mechanisms may be considered as new explicative models of collective systems. In reality, they aim to identify and, unavoidably, represent real, constituting mechanisms responsible for the emergence of the effective dynamic, phenomenological coherence.

However, the relationship of domains and selection mechanisms with the real collective phenomena is variably bidirectional, mainly due to effects of temporary, local equivalences; approximations with different consequences; and incompleteness, leading to the breaking of rules.

---

> We may intend for the system of entities to explore, perform, and define the domain configurations, allowing collective behaviors and properties in the conceptual framework of quasicoherence.
>
> ————————————
>
> The domains may give rise to other domains. For instance, other domains may be induced through the adoption of analogical, influenced behavior within noncoherent, dispersed collective entities getting in touch.
>
> ————————————
>
> What are the advantages of considering the domains of collective behaviours?
> These advantages include the possibility of avoiding an intractably high number of elements (potential infinity) by considering any combinations, variations, and extensions of properties of the domains (the interest is in the number of equivalent configurations available); the identification of characterizing evolutionary rules; outlining behaviors of collective behaviours; and setting the properties of the domain as characteristics of identity of the corresponding collective behavior together with the selection mechanism.
>
> ————————————
>
> The availability of domains and selection mechanisms should allow effective variational interventions such as the starting, varying, and deactivating of collective systems.

## 10. Further Research

Further research is expected to relate suitable profiling, scenario-making approaches based, for instance, on learning techniques, analog computation, and cognitive processing suitable to *design* domains.

Available interdependent configurational options and their contextual and evolutionary compatibility, i.e., admissibility, of domains may be, in the simplest cases, computed and designed. The available, admissible interdependence is in this case computed as dynamical intersections of local domains. However, we are dealing here with nonsimple, complex, emergent cases where coherence is not reducible to the respect of rules. Phenomenological, detected domains are algorithmically incompressible since individual, empirical data may be considered to be partly determined by causal factors and perturbations irreducible to any formal pattern. Therefore, empirical data exhibit maximal algorithmic complexity. A string is algorithmically random when it is incompressible. Depending on their perturbed nature, empirical data sets should be intended to be algorithmically random strings of digits [57,58].

In this regard, we mention the spatial or Kolmogorov complexity [59,60] which for a given system is the minimum possible length of a computer program able to entirely reproduce the behavioral features of the system.

Temporal or computational complexity is defined as the minimum time needed by a computer program, already having the minimum possible length, to reproduce the behavioral features of the system under study.

Temporal and spatial complexities can differ from each other. For instance, a deterministically (the totality of its behaviors can be generated through a very simple recursive) chaotic system has a low spatial complexity, but a very high temporal complexity (the time needed to generate a very long sequence of such behaviors is at least directly proportional to the length of the sequence itself). Accordingly, on increasing this length in an unlimited manner, the temporal complexity will also increase in an unlimited manner.

Low-complexity systems may be simulated using suitable rules-based simulators such as the flock simulator mentioned above [50].

The algorithmic incompressibility of the phenomenological complexity should be dealt with using different approaches. Data sets of real, detected phenomena should be the object of suitable ex-post approaches supposedly partially reusable to deal with other cases of comparable nature.

Further research should involve the incompressibility of domains.

This may relate to their representations and properties.

While models are intended as variables and their structures, for example, as equations which are suitable to not just imitate but represent and understand the supposed platonic essence of the phenomenon under study and allowing simulations, by contrast, nonideal models (conceptually introductory to profiles) are intended as mixtures of general approaches and specific choices—for instance, data-driven approaches that are statistically and retrospectively statistically considered or clustered.

Profiles are intended as phenomenologically data-driven, ongoing properties. Related possible approaches are given by data mining [61], which is devoted to the discovery of supposed hidden patterns in the assumption that formal regularities are hidden by their possible irregular combinations, randomness, and unsuitable scaling. Data, however, come first. Sameness for profiling is a matter of possessing the same nonanalytical properties.

Suitable approaches to be studied relate to learning [62] and profiling [63,64] techniques, analog computation [65], cognitive processing [66], and nature-inspired computation [54,56,67] suitable to identify, apply, reproduce, or generate scenarios and their properties, such as styles intended as variable recurrence and temporal or local behavioral acquisitions.

In this regard, approaches based on simplified models and profiles induce a reaction inspired by the models and profiles themselves, reducing their complexity. The effect is to reduce the complexity and introduce manipulation when the system learns to behave according to how it is considered.

Further research should have the purpose, not of reducing the complexity in the social case, but, rather, of protecting (by recognizing decomplexifying approaches) and increasing the behavioral complexity considered as the ability to deal with complex issues, such as becoming able to perform more complex tasks, e.g., dealing with the climate problem, refusing simplified, belligerent behaviors, and establishing a sustainable economy.

In sum, starting from real domains and data of collective behaviors (see Section 6.2.2), future research should then apply learning and analog computing technologies and approaches [68–70] (see Section 7.1) that are suitable for the identification of selection mechanisms for further generalizations.

Furthermore, further research should look for approaches based on the varying properties of the domains, conducts of collective behaviors, and their multiple aspects of coherence, as introduced above, to:

- Facilitate the emergence of specific collective behaviors (incubators of collective behaviors);
- Sustain ongoing collective behaviors; and
- Deactivate and disaggregate collective behaviors, such as the establishment of self-sustained collective sales and whirlpools, e.g., tornados and hurricanes. This can be a matter of establishing anticollective behaviors and using, for instance, suitable perturbations.

The concept of a domain should facilitate our understanding of processes of emergence and, most of all, our ability to influence them without using invasive, destructive, and unrealistic approaches.

The mixed usage of microscopic, mesoscopic ([4], pp. 102–128; [5]), and domain-based approaches should be better specified (is it a matter of context-dependent optimization?).

The entire proposed approach should be suitably simulated and use techniques of augmented reality [71–73].

Finally, we mention the expected generalizability of the concepts introduced above in dealing with domains of configurations of interrelated constraints when moving from the physical space to virtual spaces such as in architecture, finance, medicine, and styles, e.g., in design and music [74]. It can also be a matter of studying how domains may represent the "essential" features of Big Data [75].

When considering how fields replace objects in physics, we may consider domains to tentatively replace the materiality or the "reality" of collective interrelated materials.

See Box 13 for a summary.

**Box 13.** Summary No. 13.

---

*Further Research*

Further research is expected to relate suitable profiling, scenario-making approaches based, for instance, on learning techniques, analog computation, and cognitive processing suitable to designing domains.

Available interdependent configurational options, their contextual and evolutionary compatibility, i.e., admissibility of domains may be, in the simplest cases, computed and designed. The available, admissible interdependence is, in this case, computed as dynamical intersections of local domains.

It is a matter of using profiles intended as phenomenologically data-driven, ongoing properties. Related possible approaches are given by data mining. Data, however, come first. Sameness for profiling is a matter of possessing the same nonanalytical properties.

Suitable approaches to be studied relate to learning and profiling techniques, analog computation, cognitive processing, and nature-inspired computation suitable to identify, apply, reproduce, or generate scenarios and their properties, such as styles intended as variable recurrence and temporal or local behavioural acquisitions.

---

## 11. Conclusions

On inspiration of approaches considered by theoretical physics, such as the duality particle-field, the quantum radical understanding of particles as fields, and entanglement, we considered the possibility of introducing the representation of disperse coherent matter, such as emergent coherent behaviors, as domains. The domains are assumed to be given by achievable, admissible, subsequent, compatible, probable, possibly equivalent, and coherence-allowing composing states of configurations as authorized possible changes for entities.

The possibility of considering the coherence of collective behaviors as a property of interrelated, networked configuration domains opens the possibility of differentiating between (mathematical) properties detected in collective behaviors (such as synchronization, remote synchronization, ergodicity, long-range correlation, power laws, and self-similarity) and generative mechanisms intended to be represented by domains and processes of selection, introducing new ways to understand and, in some cases, replace the current interaction-based mechanisms. This approach is considered congenial for the multiplicity of possible states available to autonomous entities establishing collective beings, while less sophisticated approaches, e.g., energy-based optimizations and iterations, are sufficient for nonautonomous collective entities. The purpose of the new conceptual approach is to allow a move from descriptive to explanatory attempts which is finally compatible with the incompleteness and quasiness of phenomenological processes and can be used to orient, induce, and use noncomplete, corrective or prescriptive interventions [1,9].

Furthermore, domains can be used to understand quasicoherence when coherence is an inhomogeneous property that is continuously, partially recovered and contradictorily built, rather than a constraint to be respected or a formal property. Characteristics of coherent behavior, such as long-range correlation and scale freeness, should be thought of as inducible properties by suitable domains rather than as rules prescribing tout court coherence.

We differentiated between properties of collective behaviors and generative mechanisms by introducing the idea of an experimental systemics that generalizes real generative mechanisms rather than models when the real evolution or story of the system is crucial.

This allows the crucial roles of incompleteness and quasiness, sequences of non-negligible singularities, and coherence recoveries in processes of emergence, which are all analytically intractable and nonzippable in formalizations.

Because individual data may be considered to be partly determined by causal factors and perturbations that are irreducible to any formal pattern, empirical data sets should be intended to be algorithmically incompressible. Accordingly, empirical data sets exhibit maximal algorithmic complexity, and a string is algorithmically random when it is incompressible. Depending on their perturbed nature, empirical data sets should be intended to be algorithmically random strings of digits [57,58].

The mathematics of natural-inspired computation should replace well-defined complete models and properties based on simplifications often understood as essential characteristics [76].

In this regard, we considered domains of monospecies collective beings as being due to specific cognitive processing that is equal for all components. While studies on animal aggregation help to figure out the phenomenological processes at play, approaches based on domains attempt to represent some of their properties by conceptually replacing abstract models based on formal unrealistic properties that are assumed to be well-defined and constant with general validity. Domains are considered to be cognitive spaces for collective beings.

Further, the advantages of considering domains as dual of collective behaviors include the possibility of avoiding an intractable high number of elements (potential infinity) by considering any combinations, variations, and extensions of properties of the domain. This allows more realistic quasicoherence and processes of restoring and resumption to be considered, identifies characteristic evolutionary rules, outlines conducts of collective behaviors intended to be properties of their domains, and sets the properties of the domains as characteristics of the ontological identity [77] of the corresponding collective behavior.

Further research may involve the possibility of considering quantum physics coherence as being due to NG bosons.

The possible replacement of usual interaction-based mechanisms and related abstract properties through domains and selection mechanisms related to and representative of phenomenological processes opens the possibility of developing a nonclassical systemics approach to effectively deal with incompleteness, singularities, quasisystems, and quasicoherences of systems.

This, with particular interdisciplinary reference to scholars, implies the use of less idealistic approaches to complexity; requires executives to manage on the basis on domains and their properties rather than on idealistic, essential indices; and allows and requires policy makers to consider more comprehensive temporal and spatial dynamic scenarios irreducible to isolated options.

Furthermore, examples of applications relate to problems concerning disciplines such as architecture, ecology, economy, finance, medicine, and sociology, where concrete, empirical, and practical aspects cannot be ignored. Particular issues include ecosystems, the climate, and human systems with respect to economics, the environment, finance, medicine, and the use of advanced technologies (5G, 3D printers, driverless cars, robotics); social issues such as public health, pollution, public security, and safety at work, where solutions are not reducible to rules; and transversal, cross-cutting themes, such as sustainability [78]. Decisions should be considered inside a domain.

However, the conceptual approach mentioned above should not be considered as looking for the *best*, more powerful approach but, rather, for experimental, phenomenological, mixed, context-dependent usages, combining, for instance, microscopic, mesoscopic, and domain-based approaches and models. Due to their dependence on real data, the technological and computational approaches to be designed to identify domains and the selection mechanism are expected to have significant impacts on other research issues. See Box 14 for a general summary.

**Box 14.** General summary No. 14.

---

*Collective Phenomena as Quasisystems*

We mentioned how the coherence of collective phenomena is usually considered as computational, related to explicative formal models rather than due to generative mechanisms producing computational coherence as consequence and property. Formal models are usually intended to capture the essence of real phenomenological processes, material approximations or degenerations of their essence. Neglected aspects are assumed to be irrelevant and insignificant relative to the predominant essence. However, this is not the case for the emergence of collective phenomena, properly quasisystems, i.e., when a system is not always a system, not always the same system, and different coherences are active. In such a case, properties such as incompleteness, inhomogeneity, instability, irregularity, nonlinearity, incoherence-tolerance, resumptions, and restorations in balanced percentages are critical for emergence which are usually only simplified, since analytically intractable.

---

**Box 14.** *Cont.*

However, the case of collective phenomena is only representative of the complexity of a large variety of emergent phenomena, such as ecosystems, the climate, and human systems in relation to economics, the environment, finance, medicine, and social properties. It is matter of finding systemic approaches compatible with the theoretical incompleteness of complexity for nonreductionistic science, i.e., the science of emergence.

### Fields and Domains

To this end, we propose to consider approaches inspired by the concept of field, which, since electromagnetism, has replaced the classic understanding of objects in physics. In particular, we consider domains as extensions of the concept of field and characterized by the availability of multiple simultaneous options, general and local constraints, types of multiple fields, and considered dually both as generators of and generated by collective behaviours.

### Domains of Collective Entities

In particular, the domain of collective entities is not the sum or superimposition of the individual domains. Options and changes available in the domain relate to comprehensive collective systems, such as the topological; internal distribution, density; indexes of correlation; indexes of ergodicity; and statistical measures. The domain corresponds to multiple options representing possible admissible states for the entities that respect both the constraints and degrees of freedom. In sum, a domain of collective entities is assumed to represent, through (configurations of) ranges of values, the subsequent changes possible for the (collective) entity.

### Computed Domains

Domains may be computed in the case of self-tracking entities such as financial events and the collective behavior of stock price movements. Such domains allow the *profiling* of collective behaviors. Another possibility is based on considering data made available by signals generated by suitable electronic devices (GPS) and by software simulations.

### Cognitive Domains

The historical, phenomenological individual behaviors may be detected using suitable techniques such as GPS and stereometric, computer vision techniques. The sequences allow computation of the domain and fix phenomenological types of coherence.

In dealing with collective beings, i.e., entities with sophisticated cognitive systems, such as human beings, domains are, in reality, cognitive spaces for use by the entities. The behavior within cognitive spaces or cognitive domains is not computed but, rather, induced, perceived, analogically decided, remembered, imitated, attempted, and evoked. Cognitive domains may, in this case, however, be represented by some learning computations, e.g., machine learning and deep learning, which are suitable to generalise individual behaviors over time.

### Roles within Domains

A role for a generic entity is intended as simultaneous assumptions over time of different options among the ones authorized by the domains. A role should be intended as a vector of specific options, values or ranges of possible values among the ones authorized by the domains.

Domains of roles can be intended as domains of clustered available options.

Examples of roles are the simultaneous assumptions of values, among the ones authorized by the domains, for different variables, such as energetic, positional, and dynamical.

### Domains of Roles in Collective Systems

The coherence, i.e., the selection of coherent roles, is mainly energetically replicated, as for molecules in whirlpools, chemical reactions, and intended as a self-establishing convergent process, such as spontaneous synchronizations (applause, objects on vibrating surfaces, fireflies).

### The Coherence of Roles for Collective Beings

In dealing with collective beings, e.g., flocks and swarms, domains are, in reality, cognitive spaces for use by entities provided with cognitive systems. The behavior within cognitive spaces or cognitive domains is not computed but, rather:

- induced, perceived, analogically decided, remembered, imitated, attempted, evoked;

　and

- selected among options of roles as being available, compatible, continuous, possible, admissible, distinguishing between equivalent and nonequivalent, and probable.

The coherence, i.e., the selection of coherent roles, is mainly cognitive, having consequent analytical, computational aspects such as synchronization, (long-range) correlations, self-similarity, and networking. In this regard, we underline how collective beings are monospecies sharing the same cognitive system.

### The Proposed Paradigm-Shift

The conceptual paradigm-shift consists in tentatively replacing the microscopic or macroscopic interaction-based mechanisms used to understand and model collective behaviors with approaches based on domains and selection mechanisms among roles suitable to generate and maintain the integrity of quasisystems, incompleteness, incoherence tolerance, and their balances.

### Quasi-Coherence

The reason to introduce such a paradigm-shift is to allow nonidealist approaches suitable to deal with more realistic quasicoherence. Tolerance should be measured, for instance, in terms of percentages, variations, and periodicity, allowing interesting profiling information.

### Final Remarks

The conceptual approach introduced considers collective behaviors as coupled domains of instantaneous validity of configurations respecting constraints and ranges of validity for variables, together with a selection mechanism (more are possible).

What are the advantages of considering the domains of collective behaviors?

These advantages include the possibility of avoiding an intractably high number of elements (potentially infinite) by considering any combinations, variations, and extensions of properties of the domains; the identification of characterising evolutionary rules; outlining collective behaviors; and setting the properties of the domain as corresponding to a category of collective behaviors together with the selection mechanism.

The availability of domains and selection mechanisms should allow effective variational interventions such as the starting, varying, and deactivating of collective systems.

### Further Research

Further research is expected to relate to suitable profiling, scenario-making approaches based, for instance, on learning techniques, analog computation, and cognitive processing suitable to designing domains.

It is a matter of using profiles intended as phenomenologically data-driven, ongoing properties.

Suitable approaches to be studied relate to learning and profiling techniques, analog computation, cognitive processing, and nature-inspired computation suitable to identify, apply, reproduce, or generate scenarios and their properties.

This, with particular reference to scholars, implies the use of less idealistic approaches to complexity; requires executives to manage on the basis of domains and their properties rather than idealistic, essential indices; and allows and requires policy-makers to consider more comprehensive temporal and spatial dynamic scenarios irreducible to isolated options. Furthermore, examples of applications relate to problems concerning ecosystems, the climate, and human systems with respect to economics, the environment, finance, medicine, and the use of advanced technologies (5G, 3D printers, driverless cars, robotics) and social issues such as public health, pollution, public security, and safety at work not reducible to the application of rules. Decisions should be considered inside a domain.

**Funding:** This research received no external funding.

**Conflicts of Interest:** The author declares no conflicts of interest.

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
