# Peer review of "Nonclassical Systemics of Quasicoherence: From Formal Properties to Representations of Generative Mechanisms. A Conceptual Introduction to a Paradigm-Shift"

_systems, doi:10.3390/systems7040051_

Round 1
Reviewer 1 Report
I immensely liked this work that encompasses a quiet but deep epistemologic revolution. It has to do with a very old philosophical divide about what is the 'real object of investigation'.
Plato was convinced that there is a 'world of ideas' populated by the 'true images' or 'ideals' of which all the objects of this world are biased copies. It is the time-honured 'cavern metaphor' : the 'world outside the cavern' is the 'world of ideas' where the 'perfect non-noisy forms' live , the actual objects we encounter in our day-to-day life are the blurred images of such perfect forms. This attitude was at very birth of modern science and has very important (and very practical) consequences in day-to-day scientific work. It is tacitly assumed by ALL SCIENTISTS that our models are imperfect but some of them think that if we had the possibility to have a look at the 'real thing' we will discover perfect relations other scientists think the opposite: the real things are inherently fuzzy and the fact we use 'abstractions' (models) is only instrumental to our knowledge.
This second attitude provokes a completely different way of facing problems more focused on the explanation of variability than looking for invariants. Gianfranco Minati clarifies while we can face complex systems only with this last attitude and defends his thesis ina very convincing way.
Author Response
Dear reviewer,
Thank you very much for your appreciation. I really like your deep understanding of the epistemological meaning.
I attach a revised version of the article where I inserted the text highlighted in yellow to deal with the requests from the other reviewers.
Thanks again and very best regards,
Gianfranco Minati

Reviewer 2 Report
The paper aims to be a conceptual paper but, unfortunately, it is not able to do this.
I think that several limitations exist and a strong revision of the study it is necessary before of publication.
One of the weaknesses of the paper is the lacking of the methodology. I think that the author should, at least, introduce a systematic literature review to improve this study. At the end of the paper there is a research agenda that should be defined on a rigorous analysis of the literature.
Moreover, I would like to suggest to the authors to try to insert tables and figure to summarize the obtained results in order to support the full understanding.
As calimed by the author "The purpose of the article is to introduce some conceptual proposals outlining possible alternative understandings based on the concept of domain and selection mechanisms introduced afterwards that are suitable to deal with unsolved problems of Systemics, allowing less abstract and more practical approaches". I think that the paper introduce some elements but it is not able to address this issue. So I would like ot suggest to the author to revise the aim of the paper or work on the results.
Author Response
Dear reviewer,
Thank you very much for your comments and suggestions.
I attach a revised version of the article where I inserted the text highlighted in yellow to deal with the requests (I also deleted some text).
In particular I inserted
- the new Section 2 to introduce the background and the methodology;
- 13 summarizing boxes for Sections to facilitate and to support the full understanding;
- I completely revised the Section on Further research with updated literature as required;
- I accordingly updated the abstract, the Introduction, and the title emphasizing the paradigm-shift introduced.
In particular I clarified that:
Methodologically we propose for systems science a paradigmatic change equivalent to the one having occurred in Physics from object to field, namely a change from interactional entities to domains intended as extensions of fields, as it were multiple fields.
The reason to introduce such paradigm-shift is to allow non-idealistic approaches suitable to deal with the more realist quasi-coherence, when the coherence does not constantly apply to all the composing entities, but rather different forms of coherence apply.
I hope the text can be now considered suitable for publication.
Thanks again,
Gianfranco Minati

Reviewer 3 Report
see attached pdf

Author Response
Dear reviewer,
Thank you very much for your comments and suggestions.
I attach a revised version of the article where I inserted the text highlighted in yellow (I also deleted some text).
In particular I clarified that:
Methodologically we propose for systems science a paradigmatic change equivalent to the one having occurred in Physics from object to field, namely a change from interactional entities to domains intended as extensions of fields, as it were multiple fields.
The reason to introduce such paradigm-shift is to allow non-idealistic approaches suitable to deal with the more realist quasi-coherence, when the coherence does not constantly apply to all the composing entities, but rather different forms of coherence apply.
I hope this clarifies the goal of the paper.
Furthermore, I inserted:
- the new Section 2 to introduce the background and the methodology;
- 13 summarizing boxes for Sections to facilitate and to support the full understanding;
- I completely revised the Section on Further research with updated literature;
- I accordingly updated the abstract, the Introduction, and the title emphasizing the paradigm-shift introduced.
We produced a simulation software as at the reference [50] having as particularity that entities of the simulated collective behaviour are self-tracking, i.e., generating files of their positions suitable to implement the domains mentioned in the paper (Section 6.2.2).
The purpose is to implement suitable software-based domains designers to be applied to real data such as financial, medical, and social.
I hope the paradigm-shift I have in mind is not only speculative now and that the text can be now considered suitable for publication.
Thanks again,
Gianfranco Minati

Round 2
Reviewer 2 Report
Dear Author,
the paper is interesting and results are quite remarkable. The paper is improved according to the reviewer' comments and suggestions but I would like to suggest some minor revisions before publication.
Specifically, I would like to suggest to the author the following revisions:
Is it possible to summarize all the summary boxes in only one Table to insert at the end of the literature review or in the Concluding remarks? I think that the readers could appreciate a lot this table because they could focus on all characteristics analyzed and discussed in previous sections. I would like to suggest to the author to better stress implications (e.g. for scholars, organizations, policy-makers etc.) I would like to suggest to summarize the abstract, it is too long.
Finally, I would like to suggest to the author to introduce this citation:
Ponsiglione, C., Quinto, I., & Zollo, G. (2018). Regional Innovation Systems as Complex Adaptive Systems: The Case of Lagging European Regions. Sustainability, 10(8), 2862.
Author Response
Dear reviewer,
Thank you very much for your comments and suggestions.
I attach a revised version of the article where I inserted a final summarizing box highlighted in yellow (pp. 35-37). Very good idea!
At the page 34 I inserted text highlighted in yellow to deal with your request to better stress implications. In the same text I inserted the reference that you recommended.
I reduced the abstract.
I hope the text can be now considered suitable for publication.
Thanks again,
Gianfranco Minati
Reviewer 3 Report
The manuscipt has is now improved and the paper could be published in the present form.
Author Response
Dear reviewer,
Thank you very much for your acceptance to publish the paper.
However, I had to insert minor revisions (highlighted in yellow) as required by another reviewer.
Thanks again,
Gianfranco Minati